# Climatic Niche, Altitudinal Distribution, and Vegetation Type Preference of the Flea Beetle Genus *Arsipoda* in New Caledonia (Coleoptera Chrysomelidae)

**DOI:** 10.3390/insects14010019

**Published:** 2022-12-23

**Authors:** Maurizio Biondi, Paola D’Alessandro, Mattia Iannella

**Affiliations:** Department of Life, Health & Environmental Sciences, University of L’Aquila, Via Vetoio, Coppito, 67100 L’Aquila, Italy

**Keywords:** Galerucinae, Alticini, Australian region, ecological niche models, vegetation types, GIS analysis

## Abstract

**Simple Summary:**

The distribution of many beetle species remains poorly known, and the knowledge of their ecological requirements is even more fragmentary. Starting with published data on the flea beetle genus *Arsipoda* in New Caledonia, we investigated the habitat preferences of the 21 species from this area. These species are significantly associated with vegetation growing on volcanic substrates. A few widespread species are also present in secondary vegetation, such as savanna and brushwood. We estimated current suitable areas for the genus using ecological niche models, and identified possible under-sampled areas, mainly in the central sector of the main island.

**Abstract:**

New Caledonia is one of the major biodiversity hotspots. The flea beetle genus *Arsipoda* (Coleoptera Chrysomelidae) is present with 21 species, all endemic. We investigated, using GIS analyses and ecological niche models, the habitat preferences of these species in terms of vegetation types, altitude, and climate, and assessed the adequacy of knowledge on the spatial parameters affecting the distribution of the genus in New Caledonia. Altitude and geology seem to play an important role in shaping species distribution. Volcanic substrate allows the growth of ultramafic vegetation, which includes most of their host plants. From a biogeographic and conservation perspective, our results report a deep link between *Arsipoda* species and their habitats, making them particularly sensitive to environmental modifications.

## 1. Introduction

New Caledonia is noted as one of Earth’s biodiversity hotspots [1,2,3], with high levels of species richness and microendemism both of plant and animal taxa [1,4,5]. Most microendemics likely originated after the re-emergence of the Grande Terre, about 37 million years ago [6,7,8].

The New Caledonian leaf beetle fauna (Coleoptera Chrysomelidae) was first investigated comprehensively by Jolivet and Verma [9,10] and later studied with a major focus on specific subgroups [11,12,13,14,15,16,17,18,19,20].

Alticini are a tribe of leaf beetles in the subfamily Galerucinae [21]. Named ‘flea beetles’ because of a metafemoral jumping mechanism (e.g., [22]), they are small to medium size. This tribe is the largest and most diverse of Chrysomelidae, comprising over 540 genera and about 8000 extant species [23,24,25], occurring worldwide. Some genera are widespread in more than one zoogeographical region, while others are strictly endemic to very limited areas [26,27]. The highest species richness occurs in the tropics of the southern hemisphere, despite the incompleteness of our knowledge about the flea beetle fauna of those areas [25,28,29]. Adult and larval stages feed mainly on stems, leaves, or roots, and rarely on flowers, in almost all the higher plant families, generally with high specialization [30,31,32,33], and often causing economic loss when becoming invasive [34,35,36].

*Arsipoda* Erichson, 1842 is a flea beetle genus including about 90 species with distribution in Australia and New Caledonia, New Guinea and associated smaller islands, and the Solomon Islands [14,16,37,38,39,40,41,42]. This genus includes 21 species in New Caledonia, all endemic, where it is recognized as one of the notable examples of chrysomelid radiation [16,39].

The first goal of this research is to interpret the distribution of the species of *Arsipoda* in New Caledonia, in terms of habitat preference based on the presence/absence of data in the different vegetation types, as identified by Jaffré et al. [43]. Moreover, we assessed the potential habitat suitability of the *Arsipoda* genus, focusing on temperature and precipitation patterns, because of the significant influence of these parameters on the different vegetation types and, of consequence, on the distribution of these phytophagous beetles. For this aim, in order to identify possible variables contributing to limiting the suitability of the taxon considered, we built ecological niche models (ENMs) under current climatic conditions using “ensemble-modelling techniques” (see Section 2).

## 2. Materials and Methods

### 2.1. Study Area, Species Database, and Vegetation Formations

The study area encompasses New Caledonia, a French Overseas Territory located in the south–west Pacific about 1200 km east of Queensland, Australia, and 1500 km north–north–west of New Zealand. New Caledonia comprises the main island of Grande Terre, which is dominated by a chain of high mountains that run along its entire length, and several smaller islands; the Belep archipelago to the north, Loyalty Islands to the east, Île des Pins (Isle of Pines) to the south, and the Chesterfield Islands and Bellona Reefs to the west. The land area is about 19,103 km^2^.

We performed analyses on a dataset including 265 occurrence localities for the New Caledonian species of the genus *Arsipoda* (Table 1). We retrieved data records from the material examined by D’Alessandro et al. [16] and Biondi and D’Alessandro [39]. It consisted of 1550 dried pinned specimens from collections of different museums and universities worldwide [16]. The geographic coordinates for the localities (WGS84 datum) are available from the authors upon request.

All species of *Arsipoda* in New Caledonia seem to feed on pollen. They are associated with a broad range of plants; 14 plant orders and 16 plant families were recorded as possible host plants [16]: Anacardiaceae (Sapindales); Aquifoliales (Aquifoliaceae); Auracariaceae (Pinales); Cunoniaceae (Oxalidales); Cyperaceae (Poales); Dilleniaceae (incert order in APGIII System (Bremer et al. 2009)); Ericaceae, Primulaceae(now including Myrsinaceae), Symplocaceae (Ericales); Euphorbiaceae (Malpighiales); Fabaceae (Fabales); Lamiaceae (Lamiales); Myrtaceae (Myrtales); Pandanaceae (Pandanales); Proteaceae (Proteales); Winteraceae (Canellales). One species, *Arsipoda evax*, was also collected on an exotic host, the flowers of mango (*Mangifera indica*).

Regarding the vegetation types of New Caledonia, we built a raster map (cell resolution: 90 m) using the data by Jaffré et al. [43]. For the climatic aspect, we used the set of 19 bioclimatic variables available from the Worldclim.org repository, choosing the ‘current’ dataset (ver. 2.1, 30 arc-sec resolution) [44].

### 2.2. Model Building and Evaluation

To estimate the current suitable areas for *Arsipoda*, we built ENMs. We used the 19 temperature- and precipitation-related “bioclimatic” raster variables from the Worldclim.org repository, namely BIO1: annual mean temperature, BIO2: mean diurnal range (mean of monthly (max temp–min temp)), BIO3: isothermality (BIO2/BIO7) (×100), BIO4: temperature seasonality (standard deviation ×100), BIO5: max temperature of warmest month, BIO6: min temperature of coldest month, BIO7: temperature annual range (BIO5-BIO6), BIO8: mean temperature of wettest quarter, BIO9: mean temperature of driest quarter, BIO10: mean temperature of warmest quarter, BIO11: mean temperature of coldest quarter, BIO12: annual precipitation, BIO13: precipitation of wettest month, BIO14: precipitation of driest month, BIO15: precipitation seasonality (coefficient of variation), BIO16: precipitation of wettest quarter, BIO17: precipitation of driest quarter, BIO18: precipitation of warmest quarter, and BIO19: precipitation of coldest quarter. To avoid potential correlation among them, which leads to the lowering of the model’s performance, we measured both the variance inflation factor (VIF), setting the threshold = 10 [45], and Pearson’s r (|r| < 0.9, following Dormann [46] and Elith et al. [47]); for this purpose, we used the ‘vifstep’ and ‘vifcor’ functions of the ‘usdm’ R package [48]. The variables obtained as the analyses’ outcomes were then selected as predictors to calibrate the models.

We performed the models by using the entire *Arsipoda* genus occurrence localities, being aware of the drawbacks indicated by Smith et al. [49]. To account for spatial correlation among the localities, the initial dataset of 265 occurrences was rarefied through the ‘spThin’ package [50], with a minimum locality distance set to 0.1 km in R [51].

We finally built the ENMs using the “biomod2” package [52] in R environment. We generated 5 sets of 1000 pseudo-absences, and 5 evaluation runs using the “disk”, parametrizing as follows: generalized linear models (GLM): type = “quadratic”, interaction level = 3; multiple adaptive regression splines (MARS): type = “quadratic”, interaction level = 3; generalized boosting model: number of trees = 5000, interaction depth = 3, cross-validation folds = 10.

To assess the discrimination performance of the single models, we used both the area under the curve (AUC) of the ROC [47] and the true skill statistics (TSS) [53]. We used 80% of the initial dataset to build the models, and the remaining 20% for their validation. Considering the 5 evaluation runs, 5 pseudo-absences sets, and 3 modeling algorithms chosen, 75 single models were generated. Ensemble models (EMs, resulting from the combination of each ENM) were then generated for the current climatic conditions through the “BIOMOD_EnsembleModeling” function. For this purpose, we selected only the ENMs exceeding the thresholds TSS > 0.7 and AUC > 0.7 (e.g., [54,55]); we applied the “weighted mean of probabilities” (wmean), which averages the single models by weighting their AUC or TSS scores, for this purpose [52].

### 2.3. Spatial and Statistical Analyses

To assess the completeness of knowledge about the distribution of the genus *Arsipoda* in New Caledonia, that is, possible spatial bias in sampling, we measured the distance of the occurrence localities from the built-up areas. We included every building from the openstreetmap.org repository [56]. For each *Arsipoda* species, we used the full occurrence dataset. The distance raster data was generated through the ‘Euclidean distance’ tool in ArcGIS Pro 3.0 (ESRI) [57]. The ‘Presence-only Prediction’ tool in ArcGIS Pro 3.0 was applied to infer the potential presence occurrences in the study area to compare those with the “real” ones (i.e., our occurrence dataset).

To evaluate if there is a significant association between species and habitats, that is to assess whether different sets of species occur in different habitats, we performed a cluster analysis, matching occurrence localities with the vegetation types. We used the Jaccard similarity index and the weighted pair group method with arithmetic mean (WPGMA) clustering method in the NCSS-11 software. The ‘Intersect’ and ‘Extract multi values to points’ tools in ArcGIS Pro were used to extract information from the relevant spatial data used to perform and interpret the cluster analysis (vegetation types and bioclimatic variables).

## 3. Results

### 3.1. Habitat Preference, and Altitudinal Distribution

Most of *Arsipoda* occurring in New Caledonia have very limited ranges, while others, such as *A. shirleyeae* and *A. isola*, show a wide distribution within the main island (Grande Terre).

Many of the occurrence data come from areas characterized by vegetation that develops on volcanic soils, belonging to the following vegetation types (Table 2, Figure 1): “Maquis on ultramafic rock at low to middle elevation” (VT2); “Dense, humid forest on volcano sediments at low to middle elevation” (VT3); “Dense, humid forest and maquis at high elevation” (VT6); and “Dense, humid forest on ultramafic rock at low to middle elevation” (VT8). In more degraded areas, characterized by mainly secondary vegetation, such as “Savanna and secondary brushwood” (VT5) and “Swampy formations” (VT10), only *A. shirleyeae*, and above all *A. isola* were recorded, both species with a wider distribution.

The altitudinal distribution (Figure 2a) of the 21 species is rather variable. The rare *A. punctata* is known exclusively from coastal environments in the central–western region of Grande Terre (Mueo area) in “Mangroves and saline vegetation at low elevation” (VT4). High altitude (1440–1800 m) species include *A. elongata, A. reidi*, and *A. wanati*, all endemic to the Mont Humboldt area (southern Grande Terre), and *A. paniensis*, endemic to the Mont Panié area (north-eastern Grande Terre). Species with the broadest altimetric range are represented by more widely distributed elements, such as *A. agalma* (180–1500 m), *A. isola* (0–1500 m), and *A. shirleyeae* (50–1200 m). Species characteristic of medium and medium–low altitudes include *A. povilaensis* (360–450 m), endemic to the Pic D’Amoa (central–eastern Grande Terre); *A. rostrata* (500–900 m), occurring in the southern part of the main island; *A. yiambiae* (500–700 m), rare species occurring in different areas of northern and central Grande Terre; *A. atra* (700–830 m), endemic to the Aoupinié area (central Grande Terre); and *A. longifrons* (800–950 m), endemic to the Mont Humboldt area (southern Grande Terre).

Regarding the sampling effort and the possible sampling bias, a few species were collected only near to (e.g., *A. yiambiae, A. longifrons*) or far (e.g., *A. paniensis, A. atra*) from built-up areas, while others were sampled across a wide range of distances from them (Figure 2b).

The cluster analysis identified six groups (Figure 2c). The *shirleyeae–isola* cluster displays a wide habitat preference including secondary vegetation such as VT5 and VT10; *punctata* forms a group in itself, as it is the only species associated with VT4; the *elongata–reidi–wanati* group is associated exclusively with the high altitude vegetation of VT6; the clusters *transversa–longifrons* and *atra–gressitti–paniensis–povilaensis* characterize the vegetation types on volcanic substrates, VT2 and VT3, respectively; the same volcanic substrates also allows the presence of the more generalist *yiambiae–evax*; finally, the analysis returns a group of seven species associated with vegetation on ultramafic rock at low and medium altitude (VT8), among which the most significant is the *rostrata–rutai* cluster.

### 3.2. Ecological Niche Modelling and Vegetation Types

After the VIF and Pearson’s correlation analyses, we selected a set of eight uncorrelated bioclimatic variables (Figure 3) (BIO3: isothermality, range (%): 47.0–60.1; BIO7: temperature annual range, range (°C): 10.5–16.4; BIO9: mean temperature of the driest quarter, range (°C): 12.7–23.7; BIO13: precipitation of the wettest month, range (mm): 146–515; BIO14: precipitation of the driest month, range (mm): 34–217; BIO15: precipitation seasonality, range (mm): 19.8–78.9; BIO18: precipitation of the warmest quarter, range (mm): 401–1321; and BIO19: precipitation of the coldest quarter, range (mm): 165–694, which were used to calibrate the models.

After the thinning process, 173 localities were chosen for model building and calibration.

The ensemble models for the target genus result in high performance scores (AUC = 0.954 and TSS = 0.823). They return wider and more continuous suitable areas mainly in the southern region of Grande Terre (Massif du Humboldt), and more irregular and narrower suitable areas in the central (from Col des Roussetes to Col de Nassirah) and northern–eastern region (Massif du Panié) (Figure 4).

The largest areas with the lowest suitability values are in the northern part of the main island (Figure 4). Indeed, possible under-sampled areas are located in the central sector of the main island, such as the regions of Massif du Boulinda, Keiyouma, Mont Poué, and Me Maoya (Figure 5).

When coupling this data with the vegetational types, the highest suitability corresponds mainly with the ultramafic vegetation of low and medium altitude, both in humid forest and maquis formations. In the four most preferred vegetational types for *Arsipoda* in New Caledonia (VT2, VT3, VT6, and VT8) (Figure 3), the four most contributing variables to the model show different trends: BIO19 (contribution = 33.31%) assumes higher values in VT6 and VT8 and lower in VT2 and VT3, while BIO15 (16.60%) has a diametrically opposite trend; BIO13 (17.43%) has lower values in VT2 and VT8 and relatively higher values in VT3 and VT6; BIO07 (17.96%) remains quite constant.

## 4. Discussion

The sampled areas for the genus *Arsipoda* in New Caledonia do not differ much from those predicted by the ENMs. This is consistent with the distances of the occurrence localities from built-up areas: the interval is quite large, indicating extensive sampling campaigns. Moreover, notwithstanding possible bias in the collecting activity [5,16], the geographical distribution of some species groups fits the pattern of endemism identified by Pellens and Grandcolas [1].

New Caledonian *Arsipoda* species seem significantly associated with vegetation types that grow on volcanic soils: VT2 (“Maquis on ultramafic rock at low to middle elevation”), VT3 (“Dense, humid forest on volcano sediments” at low to middle elevation “), VT6 (“Dense, humid forest and maquis at high elevation”), and VT8 (“Dense, humid forest on ultramafic rock at low to middle elevation”). The preferred altitudes are medium and high, but there are also species exclusive to coastal environments, such as *A. punctata*, collected in VT4 (“Mangroves and saline vegetation at low elevation”) on the west coast of Grande Terre (Mueo area).

The habitats with more degraded vegetation have been colonized only by very few species that show a wider distribution within the main island (Grande Terre).

At the present state of knowledge, most species have very limited distributions, often corresponding to restricted high-altitude sites. This makes these species particularly vulnerable and, therefore, highly threatened.

The knowledge of the biogeography and autoecology of *Arsipoda* in New Caledonia is still incomplete. However, this flea beetle genus proves to be excellent in deepening our knowledge of the speciation and adaptation mechanisms of the New Caledonian fauna, showing possible examples of speciation due to orography-related vicariance events or adaptive divergence.

## Figures and Tables

**Figure 1 insects-14-00019-f001:**
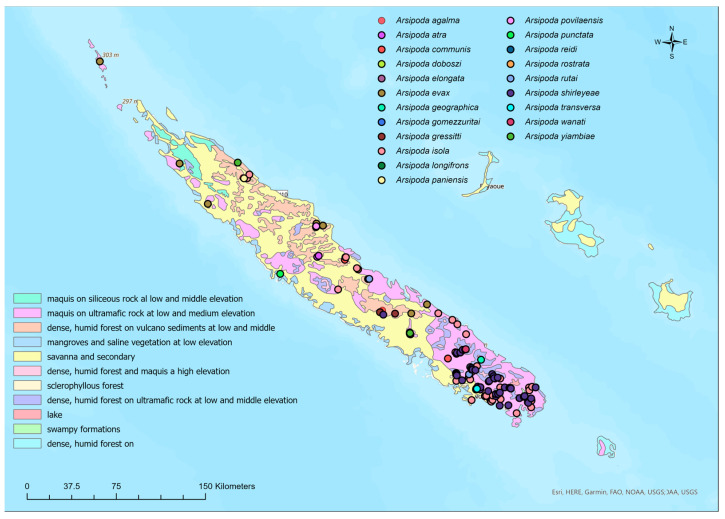
Occurrence localities for the 21 New Caledonian *Arsipoda* species and the vegetation types map redrawn by Jaffré et al. [43].

**Figure 2 insects-14-00019-f002:**
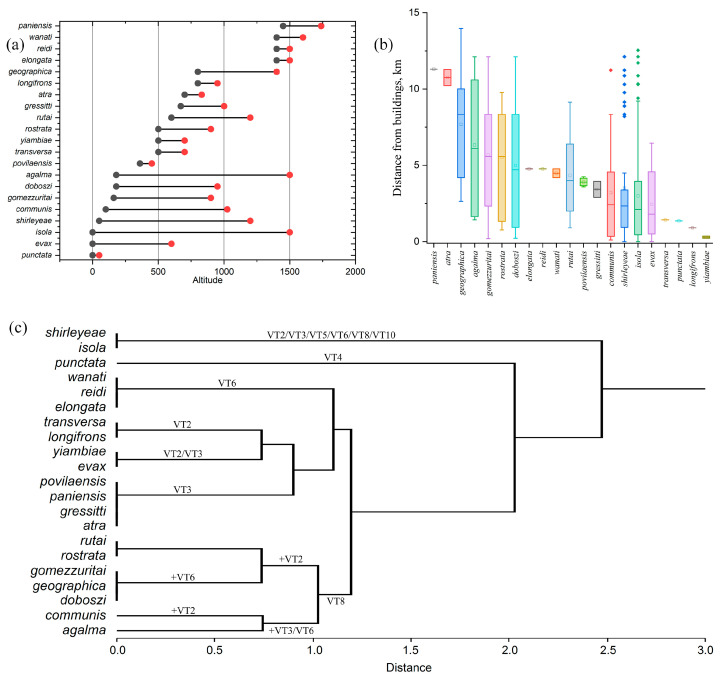
(**a**) Altitudinal ranges and (**b**) sampling distances from built-up areas for the New Caledonian *Arsipoda* species. (**c**) Dendrogram obtained from cluster analysis performed over *Arsipoda* species and their vegetation type preference.

**Figure 3 insects-14-00019-f003:**
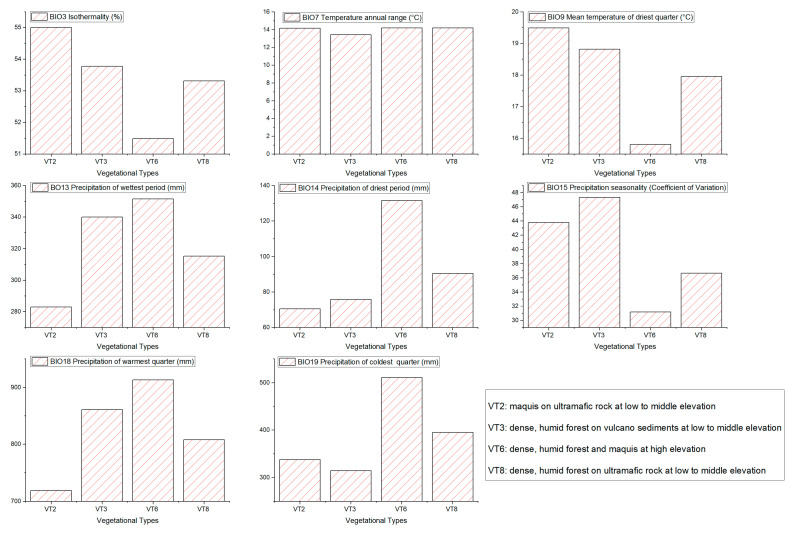
Relationship between bioclimatic variables and vegetation types of the occurrence localities of New Caledonian *Arsipoda* species.

**Figure 4 insects-14-00019-f004:**
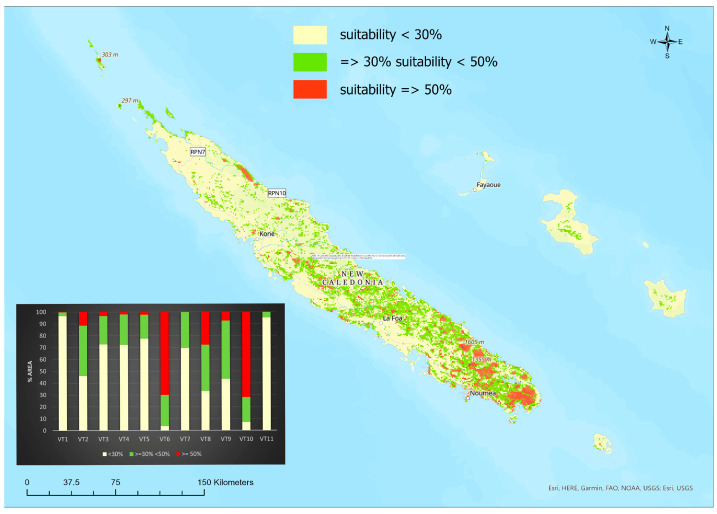
Habitat suitability of the genus *Arsipoda* in New Caledonia (map) and in each vegetation type (histogram).

**Figure 5 insects-14-00019-f005:**
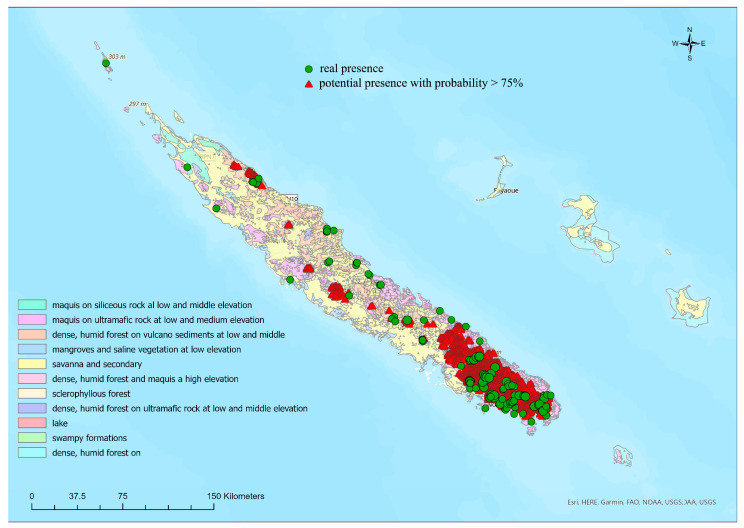
Potential/actual occurrence localities of *Arsipoda* as inferred from the ecological niche modelling process, calibrated under current climatic conditions.

**Table 1 insects-14-00019-t001:** *Arsipoda* species of New Caledonia and number of occurrences in vegetation types (following Jaffré et al. [43]) coded as follows: VT1—maquis on siliceous rock at low to middle elevation; VT2—maquis on ultramafic rock at low to middle elevation; VT3—dense, humid forest on volcano sediments at low to middle elevation; VT4—mangroves and saline vegetation at low elevation; VT5 —savanna and secondary brushwood; VT6—dense, humid forest and maquis at high elevation; VT7—sclerophyllous forest; VT8—dense, humid forest on ultramafic rock at low to middle elevation; VT9—lake; VT10—swampy formations; VT11—dense, humid forest on calcareous rock.

Species	VT1	VT2	VT3	VT4	VT5	VT6	VT7	VT8	VT9	VT10	VT11
*Arsipoda agalma* Samuelson, 1973	0	0	4	0	0	1	0	1	0	0	0
*Arsipoda atra* D’Alessandro, Samuelson, and Biondi, 2016	0	0	2	0	0	0	0	0	0	0	0
*Arsipoda communis* D’Alessandro, Samuelson, and Biondi, 2016	0	19	2	0	0	1	0	1	0	0	0
*Arsipoda doboszi* D’Alessandro, Samuelson, and Biondi, 2016	0	11	0	0	0	2	0	1	0	0	0
*Arsipoda elongata* D’Alessandro, Samuelson, and Biondi, 2016	0	0	0	0	0	1	0	0	0	0	0
*Arsipoda evax* Samuelson, 1973	0	9	4	0	0	0	0	0	0	0	0
*Arsipoda geographica* Gómez-Zurita, 2010	0	2	0	0	0	5	0	3	0	0	0
*Arsipoda gomezzuritai* D’Alessandro, Samuelson, and Biondi, 2016	0	9	0	0	0	2	0	2	0	0	0
*Arsipoda gressitti* D’Alessandro, Samuelson, and Biondi, 2016	0	0	2	0	0	0	0	0	0	0	0
*Arsipoda isola* Samuelson, 1973	0	64	1	0	11	3	0	12	0	2	0
*Arsipoda longifrons* D’Alessandro, Samuelson, and Biondi, 2016	0	1	0	0	0	0	0	0	0	0	0
*Arsipoda paniensis* D’Alessandro, Samuelson, and Biondi, 2016	0	0	1	0	0	0	0	0	0	0	0
*Arsipoda povilaensis* D’Alessandro, Samuelson, and Biondi, 2016	0	0	4	0	0	0	0	0	0	0	0
*Arsipoda punctata* D’Alessandro, Samuelson, and Biondi, 2016	0	0	0	1	0	0	0	0	0	0	0
*Arsipoda reidi* Biondi and D’Alessandro, 2022	0	0	0	0	0	1	0	0	0	0	0
*Arsipoda rostrata* Gómez-Zurita, 2010	0	6	0	0	0	0	0	1	0	0	0
*Arsipoda rutai* D’Alessandro, Samuelson, and Biondi, 2016	0	9	0	0	0	0	0	3	0	0	0
*Arsipoda shirleyeae* Samuelson, 1973	0	33	1	0	2	2	0	16	0	3	0
*Arsipoda transversa* D’Alessandro, Samuelson, and Biondi, 2016	0	1	0	0	0	0	0	0	0	0	0
*Arsipoda wanati* D’Alessandro, Samuelson, and Biondi, 2016	0	0	0	0	0	2	0	0	0	0	0
*Arsipoda yiambiae* Samuelson, 1973	0	1	1	0	0	0	0	0	0	0	0

**Table 2 insects-14-00019-t002:** New Caledonian vegetation types [43], their area, the species richness, and total number of records of *Arsipoda* species.

Vegetation Type	Area (km^2^)	No. of Species	No. of Occurrences
VT1—maquis on siliceous rock at low to middle elevation	530	0	0
VT2—maquis on ultramafic rock at low to middle elevation	4451	12	165
VT3—dense, humid forest on volcano sediments at low to middle elevation	2275	10	22
VT4—mangroves and saline vegetation at low elevation	628	1	1
VT5—savanna and secondary brushwood	8411	2	13
VT6—dense, humid forest and maquis at high elevation	132	10	20
VT7—sclerophyllous forest	16	0	0
VT8—dense, humid forest on ultramafic rock at low to middle elevation	1097	9	40
VT9—lake	23	0	0
VT10—swampy formations	43	2	5
VT11—dense, humid forest on calcareous rock	1254	0	0

## Data Availability

The occurrence localities’ dataset is available upon request; please contact the corresponding author.

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
