# Peer review of "Climatic Niche, Altitudinal Distribution, and Vegetation Type Preference of the Flea Beetle Genus Arsipoda in New Caledonia (Coleoptera Chrysomelidae)"

_insects, 2022, doi:10.3390/insects14010019_

Round 1

Reviewer 1 Report

lines 79-80: no need to mention the full name of the genus again. You can remove Erichson, 1842 (Chrysomelidae, Galerucinae, Alticini).

It is necessary to indicate in table 1, that the numbers in it are the number of registrations of the species.

Specific Comments:

1. What is the main question addressed by the research?

The main goal of the study is to identify patterns of distribution of species of the genus Arsipoda in the biocenoses of the Grande Terre and the Belep islands

2. Do you consider the topic original or relevant in the field? Does it
address a specific gap in the field?

The purpose of the study is relevant and original. Identification of patterns of distribution is extremely important for assessing the state of populations of endemic species. Many of them are endangered.

3. What does it add to the subject area compared with other published
material?

The study used museum materials and previously published data. The synthesis of these data and phytosociological data against the background of abiotic characteristics of specific habitats was performed.

4. What specific improvements should the authors consider regarding the
methodology? What further controls should be considered?

I do not see such a need.

5. Are the conclusions consistent with the evidence and arguments presented and do they address the main question posed?

The conclusions are convincing and based on the results of a correct statistical analysis.

6. Are the references appropriate?

Quoting is correct and beyond doubt.

7. Please include any additional comments on the tables and figures.

The necessary commentary on Table 1 is in the review.

Author Response

Reviewer 1

We thank Reviewer 1 for her/his comments.

Reviewer 1. lines 79-80: no need to mention the full name of the genus again. You can remove Erichson, 1842 (Chrysomelidae, Galerucinae, Alticini).

Reply. Done

Reviewer 1. It is necessary to indicate in table 1, that the numbers in it are the number of registrations of the species.

Reply. Done

Reviewer 2 Report

The authors report a thorough geographic analysis of the climatic niche characteristics of the flea beetle genus Arsipoda in New Caledonia and implications for the conservation of the genus.

It is a valuable report on the distribution of endemic species and the strong links they have with ecosystem niches.

That said, I had difficulty to follow some of the methods. Some are well details but others are succinctly listed and I am not sure how they were assembled. Also the use of terms needs to be consistent.

Remarks below:

“species until known in this area” -  correct to “species known in this area”

“allows the growth of the ultramafic vegetation” -  correct to “ allows the growth of ultramafic vegetation”

Keywords “Galerucinae Alticini” – correct to “Galerucinae, Alticini”

I don’t understand the dataset(s). They seem to be a compilation of existing data plus data collected by the authors. Was this simply pooled together? Also, the whole genus versus 21 endemics. It is not clear what is the final composition of the dataset. I gather the 21 endemics are the whole genus. It is best to only use one term to avoid confusion, i.e., 21 endemics or whole genus. Not both.

Dataset 1 (?) We performed analyses on a dataset including 265 occurrence localities for twenty-one  endemic flea beetle species of the genus Arsipoda   (Table 1).

Dataset 2 (?) We retrieved data records for the whole genus from the examined material, consisting of 1550 dried pinned specimens, and literature (see [16]), from which we recorded the geographic coordinates for the localities (WGS84 datum, available from the authors upon request).

“We included every building (as obtained  from the most recent data of the openstreetmap.org repository) to consider all the possible  facilities which could have led to a biased field sampling”

But the dataset was pinned specimens and literature.  How was this “field sampling”? you mean based on the sampling REPORTED by the literature and specimens data you have, yes?

“vegetation types and bioclimatic  variables” – bioclimatic variables are where? And where were they retrieved from? We see in Table 1 only the vegetation types.

Author Response

Reviewer 2

We thank Reviewer 2 for her/his comments.

Reviewer 2. “species until known in this area” -  correct to “species known in this area”

Reply. We changed the sentence.

Reviewer 2. “allows the growth of the ultramafic vegetation” -  correct to “ allows the growth of ultramafic vegetation”

Reply. Done

Reviewer 2. Keywords “Galerucinae Alticini” – correct to “Galerucinae, Alticini”

Reply. Done

Reviewer 2. Also, the whole genus versus 21 endemics. It is not clear what is the final composition of the dataset. I gather the 21 endemics are the whole genus. It is best to only use one term to avoid confusion, i.e., 21 endemics or whole genus. Not both.

Reply. In the Introduction, we wrote: Arsipoda Erichson, 1842 is a flea beetle genus including about 90 species with distribution in Australia and New Caledonia, New Guinea and associated smaller islands, and the Solomon Islands [14,16,37–42]. This genus includes 21 species in New Caledonia, all endemic...”

Reviewer 2. I don’t understand the dataset(s). They seem to be a compilation of existing data plus data collected by the authors. Was this simply pooled together? Dataset 1 (?) We performed analyses on a dataset including 265 occurrence localities for twenty-one  endemic flea beetle species of the genus Arsipoda   (Table 1).

Reply. Changed: “We performed analyses on a dataset including 265 occurrence localities for the New Caledonian species of the genus Arsipoda.”

Reviewer 2. Dataset 2 (?) We retrieved data records for the whole genus from the examined material, consisting of 1550 dried pinned specimens, and literature (see [16]), from which we recorded the geographic coordinates for the localities (WGS84 datum, available from the authors upon request).

Reply. Changed: “We retrieved data records from the material examined by D'Alessandro et al. [16] and Biondi and D'Alessandro [39]. It consisted of 1550 dried pinned specimens from collections of different Museums and Universities worldwide [16]. The geographic coordinates for the localities (WGS84 datum) are available from the authors upon request.”

Reviewer 2. “We included every building (as obtained from the most recent data of the openstreetmap.org repository) to consider all the possible  facilities which could have led to a biased field sampling” But the dataset was pinned specimens and literature.  How was this “field sampling”? you mean based on the sampling REPORTED by the literature and specimens data you have, yes?

Reply. Yes, we do. For this first round of revision, the sentence was we changed the sentence: “To assess the completeness of knowledge about the distribution of the genus Arsipoda in New Caledonia, that is, possible spatial bias in sampling, we measured the distance of the occurrence localities from the built-up areas. We included every building from the most recent data of the open-streetmap.org repository [56]. Occurrence localities’ dataset for each Arsipoda species was used entirety.” Based on previous information, we think the sampling clearly refers to data from D'Alessandro et al., and Biondi and D'Alessandro.

Reviewer 2. “vegetation types and bioclimatic  variables” – bioclimatic variables are where? And where were they retrieved from? We see in Table 1 only the vegetation types. 

Reply. Table 1 reported only the vegetation types. Bioclimatic variables were added in paragraph 2.2., with the repository they were retrieved from (already mentioned in 2.1.): “We used the 19 temperature- and precipitation-related “bioclimatic” raster variables from the Worldclim.org repository, namely ……..” The subset of bioclimatic variables obtained through the application of the VIF analysis, as declared in methods, were reported in paragraph 3.2.

Reviewer 3 Report

The work provides valuable new data on the ecology of endemic species of the genus Arsipoda and obviously deserves to be published.

However, I propose to supplement the text with:

-an introduction: information on the genus Arsipoda, what is known so far about this group of Chrysomelidae. The authors include only two sentences about the distribution of the species in the introduction.

-methodology: data on the time when the insects were caught and the

the method of trapping the insects and the entomological equipment used. The authors should analyse whether the manner in which the insects were trapped affected the species composition of the Arsipoda caught and, for example, the lack of Arsipoda entomofauna in some vegetation types. In habitat preference analyses, insects should be collected by a variety of methods.

Author Response

Reviewer 3

We thank Reviewer 3 for her/his comments.

Reviewer 3. I propose to supplement the text with:

-an introduction: information on the genus Arsipoda, what is known so far about this group of Chrysomelidae. The authors include only two sentences about the distribution of the species in the introduction.

Reply. This information is reported in: 16. D’Alessandro, P.; Samuelson, A.; Biondi, M. Taxonomic Revision of the Genus Arsipoda Erichson, 1842 (Coleoptera, Chrysomelidae) in New Caledonia. European Journal of Taxonomy 2016, 1–61. DOI:10.5852/EJT.2016.230. We consider including already published and easily accessible information in the text an unnecessary waste of space.

Reviewer 3. -methodology: data on the time when the insects were caught and the method of trapping the insects and the entomological equipment used. The authors should analyse whether the manner in which the insects were trapped affected the species composition of the Arsipoda caught and, for example, the lack of Arsipoda entomofauna in some vegetation types. In habitat preference analyses, insects should be collected by a variety of methods.

Reply. Our analysis is based on data collected during different field campaigns distributed over time. We think this is clear from paragraph 2.1: “We retrieved data records from the material examined by D'Alessandro et al. [16] and Biondi and D'Alessandro [39]. It consisted of 1550 dried pinned specimens from collections of different Museums and Universities worldwide [16].” Available information about collecting methods is reported in: 16. D’Alessandro, P.; Samuelson, A.; Biondi, M. Taxonomic Revision of the Genus Arsipoda Erichson, 1842 (Coleoptera, Chrysomelidae) in New Caledonia. European Journal of Taxonomy 2016, 1–61. DOI:10.5852/EJT.2016.230.